

# Vegetation dynamic analysis based on multisource remote sensing data in the east margin of the Qinghai-Tibet Plateau, China

Haijun Wang[1,2], Peihao Peng[1], Xiangdong Kong[2,3], Tingbin Zhang[1] and Guihua Yi[1]

[1] College of Earth Science, Chengdu University of Technology, Chengdu, Sichuan, China
[2] Engineering and Technical College of Chengdu University of Technology, Leshan, Sichuan, China
[3] School of Civil Engineering and Architecture, Southwest Petroleum University, Chengdu, China

## ABSTRACT

This study focuses on the vegetation dynamic caused by global environmental change in the eastern margin of the Qinghai-Tibet Plateau (EMQTP). The Qinghai–Tibet Plateau (QTP) is one of the most sensitive areas responding to global environmental change, particularly global climate change, and has been recognized as a hotspot for coupled studies on changes in global terrestrial ecosystems and global climates. An important component of terrestrial ecosystems, vegetation dynamic has become a key issue in global environmental change, and numerous case studies have been conducted on vegetation dynamic trends using multi-source data and multi-scale methods across different study periods. The EMQTP is regarded as a transitional area located between the QTP and the Sichuan basin, and has special geographical and climatic conditions. Although this area is ecologically fragile and sensitive to climate change, few studies about vegetation dynamics have been carried out in this area. Thus, in this study, we used long-term series datasets of GIMMS 3g NDVI and VGT/PROBA-V NDVI to analyze the vegetation dynamics and phenological changes from 1982 to 2018. Validation was performed based on Landsat NDVI and Vegetation Index & Phenology (VIP) data. The results reveal that the year 1998 was a vital turning point in the start of growing season (SGS) in vegetation ecosystems. Before this turning point, the SGS had an average slope of 9.2 days/decade, and after, the average slope was 3.9 days/decade. The length of growing season (LGS) was slightly prolonged between 1982 to 2015. Additionally, the largest national alpine wetland grassland experienced significant vegetation degradation; in autumn, the degraded area accounted for 63.4%. Vegetation degradation had also appeared in the arid valleys of the Yalong River and the Jinsha River. Through validation analysis, we found that the main causes of vegetation degradation are the natural degradation of wetland grassland and human activities, specifically agricultural development and residential area expansion.

Corresponding authors
Haijun Wang,
wanghaibo.2006@163.com
Peihao Peng, peihaop@163.com

## INTRODUCTION

Vegetation is the main component of terrestrial ecosystems on earth (*Piao & Fang, 2003*), playing a crucial role in energy exchange, water cycles and biological cycles on the terrestrial surface. Vegetation connects the atmosphere, hydrosphere and biosphere, and is vital for reducing greenhouse gases, regulating carbon balance, and maintaining climate stability at the global scale (*Hu et al., 2010*). Vegetation dynamics are especially sensitive to climate change (*Yang & Piao, 2006*) and, in recent years, have been regarded as the key element in the global changes of terrestrial ecosystems (*Kelly, Tuxen & Stralberg, 2011*; *Li et al., 2019a*; *Li et al., 2019b*).

Because of its long-term series span and large area coverage, remote sensing data has become a common and vitally important data source in the field of vegetation dynamics monitoring (*Casa et al., 2018*). The normalized difference vegetation index (NDVI) is widely used as a parameter of vegetation dynamics and refers to the quantitative values of vegetation conditions. It is often obtained through a combination of different spectral remote sensing data. Due to its sensitivity to vegetation growth status, phenology changes, and vegetation cover change (*Tucker, 1979*), NDVI has been the most widely applied indicator used to represent vegetation status among various other vegetation indices. Global Inventory Monitoring and Modeling Studies (GIMMS) 3g NDVI and SPOT VGEATION (VGT) NDVI have been acknowledged as the most widely used remote sensing data products (*Casa et al., 2018*; *Chavez et al., 2019*), being used to monitor vegetation dynamics at both regional and global scales (*Liu et al., 2017a*; *Liu et al., 2017b*; *Wu et al., 2017*). GIMMS NDVI data cover a large time span from 1982 to 2015, and VGT NDVI has higher spatial and temporal resolutions.

One of the most sensitive areas responding to global environmental change (*Yang & Piao, 2006*; *Li et al., 2019a*; *Li et al., 2019b*), the QTP has been recognized as a hotspot for vegetation dynamics. *Peng et al. (2012)* used the Hurst exponent to analyze the trend of vegetation dynamics, and an eco-environmental vulnerability change was performed in the Sanjiangyuan region of the QTP based on Liu's fuzzy analytic hierarchy process (*Liu et al., 2017a*; *Liu et al., 2017b*). Shen and colleagues looked at the spring vegetation phenology change in the QTP in the last decade (*Shen et al., 2014*), and the response of artificial vegetation to changes in the ecological environment of the QTP was studied by *Li et al. (2017)*. The EMQTP is a transitional zone reaching from the Sichuan basin to the QTP (*Li et al., 2019a*; *Li et al., 2019b*; *Amelie et al., 2009*). It belongs in the Qinghai-Tibet Plateau climate zone and is affected by southeast monsoons. The largest alpine wetland and natural pasture are located in the EMQTP, and alpine and subalpine meadows are the dominant vegetation in this area. Additionally, this area contains the important Yangtze and Yellow Rivers. This area has special conditions that make it suitable for the research of terrestrial ecosystems, particularly wetland changes and vegetation dynamics. The terrestrial ecosystem of the QTP has also transformed (*Li et al., 2019a*; *Li et al., 2019b*; *Fei et al., 2018*) with the current backdrop of global climate change *IPCC, 2014*. How has the terrestrial ecosystem, specifically the vegetation, of the EMQTP (a transitional and special area) changed? The answer to this question has significant meaning for the ecological

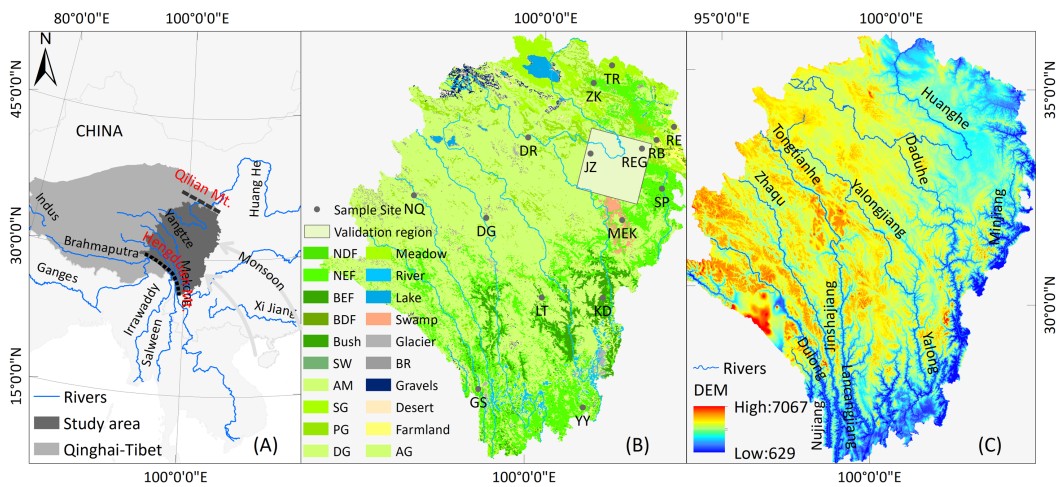

**Figure 1** Geographical location (A), land use and land cover (B, GLC2015), and the digital elevation model (C, STRM-30 m) of the study area.

protection of the river source area. However, few studies have focused on this area for vegetation dynamic research. Previously, Peng analyzed vegetation change trends from 1982 to 2001, and tested the results by *P* values in the QTP area (*Peng et al., 2012*). This was a well-executed study, but has two vital problems that need to be discussed: first, the *P* value test may be unreliable since it is easy to pass (*Woolston, 2015*), necessitating the reevaluation of the significance of the vegetation change trend; second, the NDVI statistics for the various vegetation were not described clearly. Were statistics taken at a site scale or at a regional scale? This ambiguity brings into question the validity of the results.

In this work, we studied the EMQTP (a sub-area of the QTP) (Fig. 1) using multisource NDVI images and a semi-innovative approach to analyze vegetation dynamic trends between 1982 to 2018. More specifically, this study attempted to address the following questions:

(1) What was the multi-scale phenological change of the study area?

(2) What were the temporal-spatial dynamics of different vegetation?

## MATERIALS & METHODOLOGIES

### Study area

The EMQTP (Fig. 1) extends from Hengduan Mountain to Qilian Mountain (26.1–36.2°E, 94.0–104.4°N). It includes the Ganzi, Aba autonomous prefecture of Sichuan province, and parts of Qinghai Province and Tibet. This is the source region for some important rivers, such as Yangtze River and the Yellow River. Moreover, this region is the largest natural pasture and has the largest alpine wetland grassland in China, and alpine and sub-alpine meadows dominate this area. Frequent human activities and climate changes in this area have caused transformations in the vegetation cover and degradation to the grassland. This area has been defined as a typical ecologically fragile area by the government, and accordingly needs to be researched and protected.

## Image and processing
### Image sources

*(1) GIMMS 3g NDVI dataset.* The most recent version of the Global Inventory Modeling and Mapping Studies (GIMMS) 3g NDVI (*NASA-NEX, 2019*) (January 1982 to December 2015) by AVHRR sensor was the dataset used in this study. The 3g data were produced in a geographical coordinate with a 15-day interval and a spatial resolution of 0.083° × 0.083° per pixel, and had been submitted to atmospheric and radiometric correction. The 3g NDVI data were generated to improve the reliability for areas with short growing seasons (*Zheng et al., 2017*; *Florian et al., 2016*; *Du et al., 2015*). The 3g NDVI data processed using MVC methods generated monthly NDVI values. This process eliminated the effects of atmosphere, clouds, soil, and snow. The 3g data were downloaded from ECOCAST (https://ecocast.arc.nasa.gov).

*(2) SPOT VGT \PROBA-V NDVI dataset.* Other long-term NDVI series datasets used in this study were the SPOT VGT (S10) and PROBA-V, a 10-day composite NDVI dataset taken at a spatial resolution of 1 × 1 km from the period of April 1998 to October 2018. The quality of the VGT NDVI dataset in terms of geometry and radiometry processing for directional and atmospheric effects marks it as an excellent instrument for monitoring forests, grasses, and crops (*VITO, 2018a*). Additionally, the PROBA-V NDVI (*PROBA-V Products User Manual, 2018*) data at 1 km resolution were also used in this study to extend the monitoring time. A PROBA-V satellite was launched in May 2013 to plug the gap between SPOT-VGT and Sentinel-3 satellites. The two NDVI datasets were generated using the MVC method (*Holben, 1986*), as MVC was confirmed as a reliable and practical method from many previous studies. The VGT\PROBA-V NDVI dataset has been used widely in forest, grassland and crop dynamic studies largely due to its high quality (*PROBA-V Products User Manual, 2018*). The data were downloaded from VITO Earth Observation (http://www.vito-eodata.be).

*(3) Data Engine of VIP Lab.* The data engine of the VIP (Vegetation Index & Phenology) Laboratory provides phenology data at a global range using a three-dimensional grid. The VIP product (*VIP Lab, 2011*) was generated based on the AVHRR (Advanced Very High-Resolution Radiometer) dataset with a spatial resolution of 0.05° × 0.05° per pixel. This is a yearly phenology product with a time span from 1982 to 2015. This product includes the start of season, end of season, and the day of peak. VIP phenology products are available from the Vegetation Index & Phenology Lab (https://vip.arizona.edu).

*(4) Landsat NDVI.* Landsat Thematic Mapper (TM) and Operational Land Imager (OLI) data were used to validate the NDVI-based (GIMMS 3g and VGT\PROB-V) vegetation dynamic trends. The Landsat 5 (L5) image consists of seven bands with a spatial resolution of 30 m. Band 3 (0.63–0.69 µm) is in the red (R) spectrum, and band 4 (0.76–0.90 µm) is near the infrared red (NIR) spectrum. Landsat 8 (L8) OLI includes nine bands with a spatial resolution of 30 m. Among them, Band 4 is in the R spectrum (0.63–0.68 µm) and Band 5 is in the NIR spectrum (0.845–0.885 µm). We used NIR and R band to calculate NDVI, and validation was performed based on this vegetation index. The study area is
covered by a large number of clouds, especially in summer, and we needed to select images with less than 5% clouds from the USGS website. The basic information of the images downloaded, including the orbital number, sensor type, and image acquisition date, are shown in Table 1.

*(5) Vegetation and topographic maps.* The vegetation vector map is at 1: 100000 scale. Global Land Cover 2015 raster data and terrain data (SRTM-DEM) of the study area have a spatial resolution of 30 m, and they are available from the USGS database (https://earthexplorer.usgs.gov/).

### Data processing

*Step 1. Smoothing processing of NDVI images.* A simplified least squares-fit convolution was proposed by Savitzky and Golay in order to smooth and compute derivatives of a set of consecutive spectrum values (*Cao et al., 2018*; *Savitzky & Golay, 1964*). The convolution can be recognized as a weighted moving average filter, with weight allocated as a polynomial of a certain form. We can apply the filter to any consecutive data when data points are at a fixed and uniform interval along the chosen abscissa, and the curves formed by graphing the points must be continuous and smooth. The long-term series GIMMS 3g NDVI and SPOT VGT NDVI just satisfy these conditions. The general equation of the simplified least-squares convolution for NDVI time-series smoothing is as follows:

$$NDVI_j^* = \frac{\sum_{i=-m}^{i=m} C_i NDVI_{j+i}}{N}, \tag{1}$$

Where $NDVI$ is the raw pixel value, $NDVI^*$ is the pixel smoothed value, $C_i$ is the coefficient for the $i$th NDVI value of the filter, and $N$ is the number of convoluting integers. It is equal to the smoothing window size (2m + 1). The index $j$ is the running index of the original ordinate data table. The smoothing array (filter size) consists of 2m + 1 points, where $m$ is the half-width of the smoothing window. The coefficients of a Savitzky–Golay filter ($C_i$) are directly available from the latest version (*Madden, 1978*; *Savitzky & Golay, 1964*; *Steinier, Termonia & Deltour, 1972*).

*Step 2. Co-registration of NDVI images.* The coordinates of the downloaded raw images showed some deviations because they were imaged with different sensors. In order to better analyze the data, especially the four types of images of the same area, we needed to make a geometric co-correction. In light of the large difference in the spatial resolution of images, the co-registration method based on image-to-image was performed for GIMMS 3g NDVI, VGT/PROBA-V, and Landsat NDVI. We selected the same geo-features, such as rivers, lakes, and mountains, to correct the images. The accuracy of this correction method is higher than the point-to-point method based on ground control points (GCP).

*Step 3. Sampling and statistics.*

*(1) Image sampling.* Sampling is indispensable when finding the trend analysis of NDVI at the site scale. The location and the size of the sample especially will impact the results

Wang et al. (2019), *PeerJ*, DOI 10.7717/peerj.8223
**Table 1  Acquisition date and sensor of Landsat orthophoto used in this study.**

| Path/Row | 131/37 | | | | | | | | | |
|---|---|---|---|---|---|---|---|---|---|---|
| Sensor | Landsat 5 TM | | | | | | | | Landsat 8 OLI | |
| Date (y/m/d) | 1999/09/29 | 2000/10/04 | 2001/10/10 | 2003/09/28 | 2005/10/03 | 2007/09/27 | 2008/10/03 | 2011/10/06 | 2013/10/06 | 2015/10/01 |
| Resolution | 30 m | | | | | | | | | |

**Table 2  Vegetation types and their use of the sample site.**

| Site | Longitude (°E) | Latitude (°N) | Altitude (m) | Vegetation type | Data use |
|---|---|---|---|---|---|
| Songpan (SP) | 103.567 | 32.650 | 3732 | Needle-leave deciduous forest (NEF) | Phenology analysis and vegetation dynamic |
| Ruobei (RB) | 103.894 | 34.105 | 3155 | | |
| Zeku (ZK) | 101.504 | 35.094 | 3889 | Broadleaved deciduous forest (BDF) | Vegetation dynamic |
| Gongshan (GS) | 98.667 | 27.750 | 2297 | | |
| Yanyuan (YY) | 101.517 | 27.433 | 2571 | | |
| Kangding (KD) | 101.967 | 30.050 | 4038 | Broadleaved evergreen forest (BEF) | Vegetation dynamic |
| Litang (LT) | 100.267 | 30.000 | 4581 | | |
| Jiuzhi (JZ) | 101.483 | 33.433 | 3985 | Alpine meadow (AM) | Phenology analysis and vegetation dynamic |
| Drerong (DR) | 99.650 | 33.750 | 4396 | | |
| Nangqian (NQ) | 96.483 | 32.20 | 3661 | | |
| Tongren (TR) | 103.389 | 33.796 | 3164 | Typical meadow (TM) | Vegetation dynamic |
| Dege (DG) | 98.583 | 31.800 | 3918 | | |
| Ruoergai (REG) | 102.967 | 33.583 | 3444 | Plain grassland (PG) | Vegetation dynamic |
| Maerkang (MEK) | 102.451 | 31.896 | 4708 | Swamp (SW) | Vegetation dynamic |
| Ruoe (RE) | 102.017 | 35.517 | 3229 | Farm land (FL) | Phenology analysis and vegetation dynamic |

of NDVI trend analysis. Many factors were considered, such as the sample being an area of vegetation cover, and only using the single type of vegetation. Moreover, mixed pixels were eliminated for NDVI data with two different resolutions. Using the analysis above, we selected 15 representative sample sites, with a sample size of 8 km × 8 km (GIMMS 3g NDVI, 1 × 1 pixel. VGT\PROBA-V NDVI, 8 × 8 pixels). The 15 sample sites selected were Songpan (SP), Kangding (KD), Litang (LT), Jiuzhi (JZ), Derong (DR), Nangqian (NQ), Tongren (TR), Dege (Dege), Ruoergai (REG), Maerkang (MEK), Ruobei (RB), Zeku (ZK), Gongshan (GS), and Yanyuan (YY). The location and vegetation types of each site are shown in Table 2 and Fig. 2.

*(2) Sample statistics.* The basic principle of the sample statistics (Fig. 3) is that the calculation of pixel parameters was performed by setting a zonal window with a fixed size. The calculated parameters included the sum, mean, maximum, and minimum of the pixel values in the zonal window. The size of the zonal window was set to 8 km × 8 km according to the lower resolution of the two different NDVI images in this study. The results were divided into three cases (possibilities) for GIMMS 3g images, using the mean statistics of a GIMMS 3g image as an example. The formulations are as follows.
*(2) Sample statistics.*

$$\text{Case}_1 : \text{Mean} = V_1 \ or \ V_2 \ or \ V_3 \ or \ V_4, \tag{2}$$

$$\text{Case}_2 : \text{Mean} = \frac{V_1 + V_2}{2} or \frac{V_1 + V_3}{2} or \frac{V_2 + V_4}{2} or \frac{V_3 + V_4}{2}, \tag{3}$$
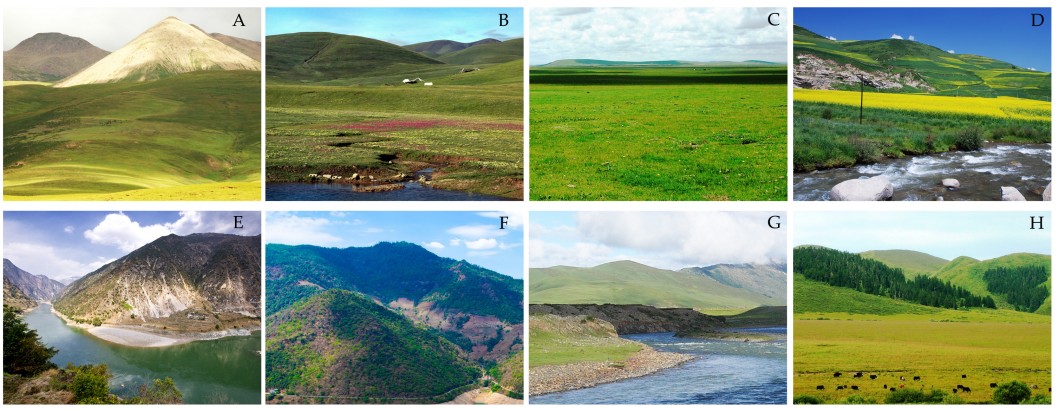

**Figure 2** **Vegetation cover types of sample site in study area.** (A) NQ, (B) DG, (C) REG, (D) TR, (E) GS, (F) YY, (G) LT, and (H) SP. Photo credit: Haijun Wang.

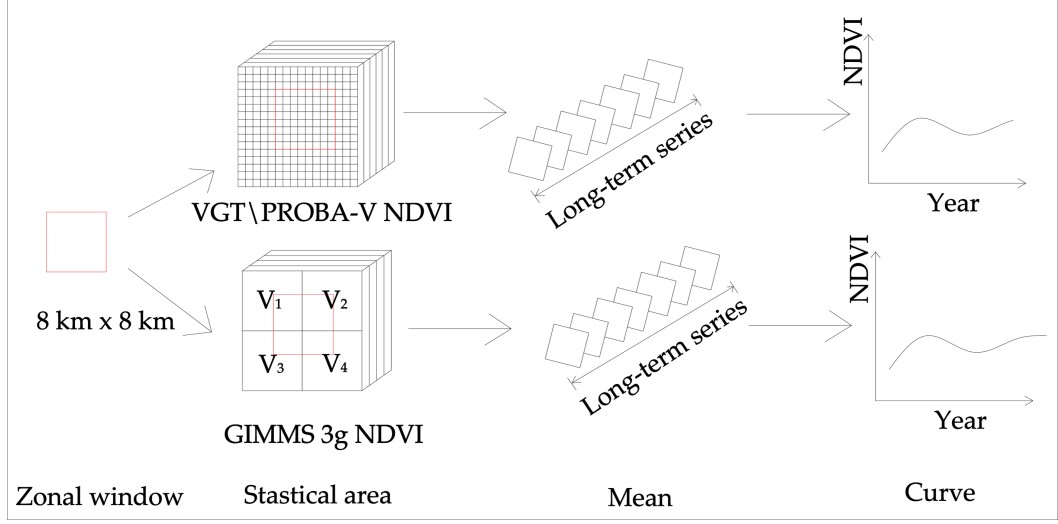

**Figure 3** **Schematic diagram of sample statistics.**

$$\text{Case}_3 : \text{Mean} = \frac{V_1 + V_2 + V_3 + V_4}{4}, \tag{4}$$

Where $V$ is the pixel value of GIMMS 3g NDVI images. According to this principle, the formula above was additionally followed for VGT/PROBA-V NDVI image statistics. Due to its higher resolution, more pixels were counted than in GIMMS NDVI images. Compared to the Point to Value Method (PVM), this method provides more accurate statistical results and reduces invalid values.

## Methodologies

### Detection of phenological change

Previous studies (*Chandola et al., 2010*; *Chu et al., 2018*; *Gu et al., 2018*; *Tang et al., 2018*) have used the dynamic threshold method to define the start of the growing season (SGS), the end of the growing season (EGS), and the length of the growing season (LGS). This method defines SGS and EGS as the moment when the pixel values increase and decrease, respectively, to a certain proportion of NDVI amplitude in one year. This method can avoid the mutual interference caused by different regional water-heat conditions, vegetation, and soil types in the fixed threshold method. The specific formulation is as follows:

$$D_{SGS,EGS} = (NDVI_{max} - NDVI_{min}) \times C, \tag{5}$$

Where $NDVI_{max}$ is the maximum value and $NDVI_{min}$ is the minimum value of NDVI in one year. $C$ is the threshold ratio. According to previous studies (*Yu, Luedeling & Xu, 2010*; *Jonsson & Eklundh, 2002*) and the annual variation of vegetation in the study area, the SGS and EGS thresholds were set at 20% in the sample site. $D$ is a time judgment function where the time point is taken as EGS or SGS when NDVI increases or decreases to the threshold $C$. The time span from SGS to EGS is the LGS.

### Trend analysis of the NDVI change

The stability and accuracy of the trend analysis method have been tested and it is widely used (*Chu et al., 2018*; *Gu et al., 2018*; *Tang et al., 2018*). This method was applied at a pixel scale in the period 1982–2018 to detect NDVI change and distribution. The formula is:

$$slope = \frac{n \times \sum_{i=1}^{n}(i \times NDVI_i) - \sum_{i=1}^{n} i \sum_{i=1}^{n} NDVI_i}{n \times \sum_{i=1}^{n} i^2 - (\sum_{i=1}^{n} i)^2}, \tag{6}$$

where $i$ is the number of the year, with the range 1 to $n$ (GIMMS NDVI, $n = 34$; VGT NDVI, $n = 20$). $NDVI_i$ represents the growing season NDVI of year $i$. The slope is the trend in NDVI from 1982 to 2018. A *slope* >0 indicates an increasing trend, while a *slope* <0 represents a decreasing trend during the study period.

### Mann-Kendall trend test

The Mann-Kendall (M-K) trend test has been widely used as a non-parametric statistical test method (*Gocic & Trajkovic, 2013*; *Yavuz, 2018*). This test method does not require samples for normal distribution, and it has a high degree of quantification. This method is suitable to trend test time series data, especially the data of long-term series. The principles of the M-K trend test are as follows.

$$S_k = \sum_{i=1}^{k} r_i, k = 2, 3 \cdots, n \tag{7}$$

$$r_i = \begin{cases} +1 \ x_i > x_j \\ +0 \ x_i < x_j \end{cases} \quad j = 1, 2 \cdots, i \tag{8}$$

Where $n$ is the number of the sample, and $x$ is the time span. $s_k$ is the cumulative number of values at the time $i$, greater than those at the time $j$.

$$UF_k = \frac{|S_k - E(S_k)|}{\sqrt{var(S_k)}}, k = 1, 2 \cdots, n \quad (9)$$

$$\begin{cases} E(S_k) = \dfrac{k(k-1)}{4} \\ var(S_k) = \dfrac{k(k-1)(2k+5)}{72} \end{cases}, k = 2, 3, \ldots, n \quad (10)$$

Where $UF_k$ is a statistic calculated by $x_1, x_2, \ldots, x_n$. $E(S_k)$, and $var(S_k)$ is the mean and variance of $S_k$. $UB_k = -UF_k (k = n, n-1, \ldots, 1)$, $UB_1 = 0$. $a$ is the signification level. If $a = 0.05$, then the $u_{0.05} = \pm 1.96$.

A $UF_k > 0$ indicates an increasing trend, while a $UF_k < 0$ represents a decreasing trend. When the critical line is exceeded ($u_{0.05} = \pm 1.96.$), a significant upward or downward trend is shown.

### Bayesian trend test

The traditional approach of trend analysis cannot detect the turning point for a set of data in a time series, and cannot estimate the change after it happens. The Bayesian method can overcome these difficulties. The Bayesian theorem constructed the probability distribution (posterior probability) function of the turning point of data change trends. The specific formula is:

$$P(\theta | \text{data}) = \frac{P(\text{data} | \theta) \times P(\theta)}{P(\text{data})}, \quad (11)$$

Where $P(\theta)$ denotes the prior probability distribution. $P(\theta j \text{data})$ is the posterior probability distribution, representing the confidence level of a turning point of change trends. $P(\text{data})$ indicates the evidence; more importantly, it indicates the moment of data peak. For example, in a set of data in time series $X = \{x_{1982}, x_{1983}, x_{1984} \cdots x_{2015}\}$, how do the data change? We would need to calculate the moment and the probability of the turning point occurring when the trend changes. We referred to previous studies (*Hèou et al., 2017*; *Peter et al., 2018*; *Zhao et al., 2019*; *John, Thomas & Christopher, 2015*) to calculate the moment and probability of the turning point occurring (Fig. 4). The combination of the Bayesian approach and M-K trend analysis can improve the accuracy of trend analysis, and can validate M-K trend analysis. The flowchart of methodology used in this study is shown in Fig. 5.

## RESULTS

### GIMMS 3g NDVI-based vegetation dynamic trends from 1982 to 2015

#### Phenological trends of vegetation

The dynamic threshold method was used to detect the phenological changes of three ecosystems from 1982 to 2015. The trend of phenology change (Figs. 6A, 6B, 6C, 6G, 6H,

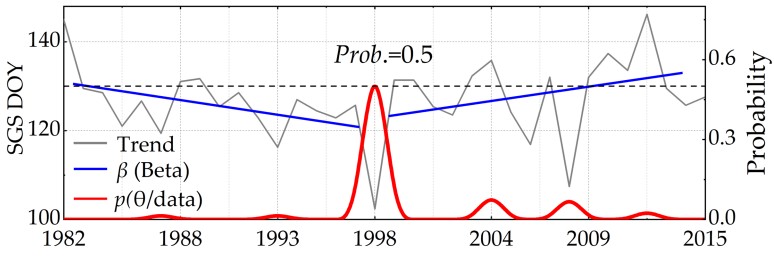

**Figure 4** **The trend and the confidence level of NDVI change based on the Bayesian method.** Beta ($\beta$) represents the continuous probability of trend change.

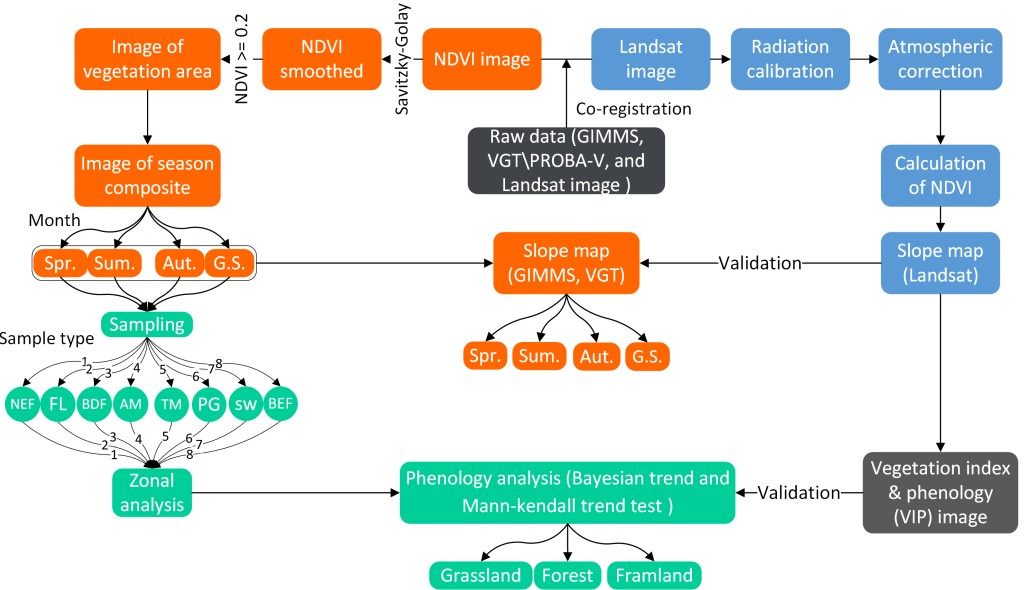

**Figure 5** **Flowchart of methodology used in this study.**

and 6I) and the M-K trend test (Figs. 6D, 6E, 6F, 6J, 6K, and 6L) are shown in Fig. 6. In the forest ecosystem (6A and 6E), the SGS change trend rose (*slope* = 10.08 days/decade, *UF* <0, *Sig.* = 0.05), and then began to fall (*slope* = 2.06 days/decade, *UF* >0, *Sig.* = 0.05), showing overall trending delays (*slope* = 0.66 days/decade). The EGS showed a slight decline from 1982 to 2015 (*slope* = 0.15 day/decade), although it began to advance after 1998 (*slope* = 0.89 day/decade). It was calculated that the LGS decreased by 0.51 days/decade based on the SGS and EGS during the study period. The curves of the SGS and EGS for grassland ecosystems are shown in Figs. 6C and 6I. The SGS of the grassland ecosystem was accelerated by 0.64 days/decade, and the EGS was delayed by 0.46 days/decade. Accordingly, the LGS was prolonged by 1.1 days/decade from 1982 to 2015. The variation trend of SGS in the grassland ecosystem was consistent with that of the forest ecosystem. The year 1998 was the turning point of the curve, and after that year, the SGS curve showed trending delays (*slope* = 2.68 days/decade). For the farmland ecosystem

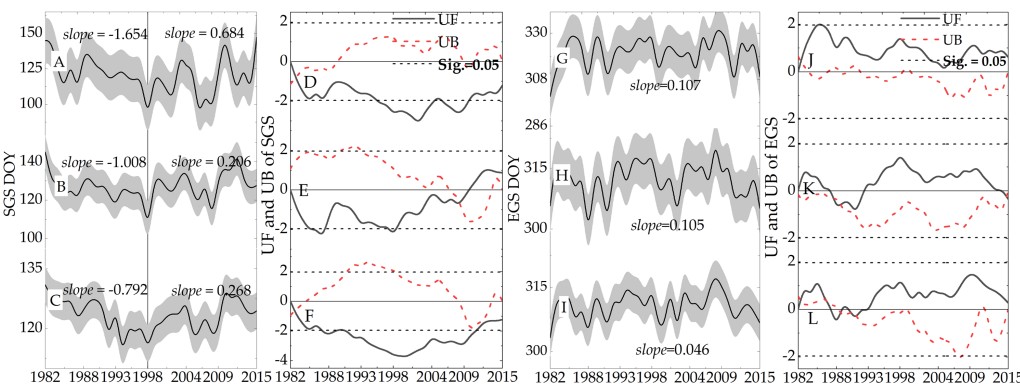

**Figure 6** **Trend and test of SGS and EGS from 1999 to 2018.** The gray background area is the standard deviation (SD). (A) Farmland. (B) Forest. (C) Grassland. (D) Farmland. (E) Forest. (F) Grassland. (G) Farmland. (H) Forest. (I) Grassland. (J) Farmland. (K) Forest. (L) Grassland.

(Figs. 6A and 6G), the trend of change was consistent with that of the grassland and forest ecosystem, except for the change rate of SGS. From 1982 to 2015, the SGS trend was accelerated by 2.98 days/decade, and the EGS was delayed by 1.07 days/decade. Due to the acceleration of the SGS and the delay of the EGS, the LGS was prolonged by 8.74 days/decade (1982-2015).

### Spatial dynamics of vegetation

We calculated the trend and spatial distribution of NDVI changes across different seasons from 1982 to 2015 (Fig. 7). In spring (Fig. 7A), the trend range of NDVI change was from -0.0679/decade to 0.0693/decade. Vegetation increased most in the area with a percentage of 82.1%, and were mainly distributed in SP, GS, YY, REG and MEK. The needle-leaf deciduous forest showed vigorous activity in SP, GS, and YY, and the vegetation coverage of swamp and plain grassland also experienced an upward trend in REG and MEK. The vegetation dynamics showed a weak upward trend in DR, JZ, and NQ. The slope was 0.005/decade to 0.013/decade in the alpine and subalpine meadows distributed in this area. In addition, vegetation deterioration had occurred in regions of the EMQTP, such as the west of ZK and DR. The west of DR was especially affected, as it is a desert grassland with low vegetation coverage. As shown in Fig. 7B, summer vegetation experienced great degradation with more than 51.5% of the area showing a downward trend. In particular, the south section of the study area had a slope of 0.08 / decade. The proportion of the vegetation increase area was 71.7%, and was mainly distributed in NQ and DG regions due to the degeneration of desert grassland. The vegetation dynamic in autumn (Fig. 7C) was basically consistent with the mean trend of the full growing season (Fig. 7D). The area of NDVI increase accounted for 75.1% in the full growing season. The difference between autumn and the growing season is only shown in the slope rate of NDVI increase and decrease. In the growing season, the slope was -0.0394/decade to 0.0486/decade. The maximum value appeared in RE, the south of SP, and the central region of the study area. Through the analysis of vegetation dynamics across different seasons, we found that the

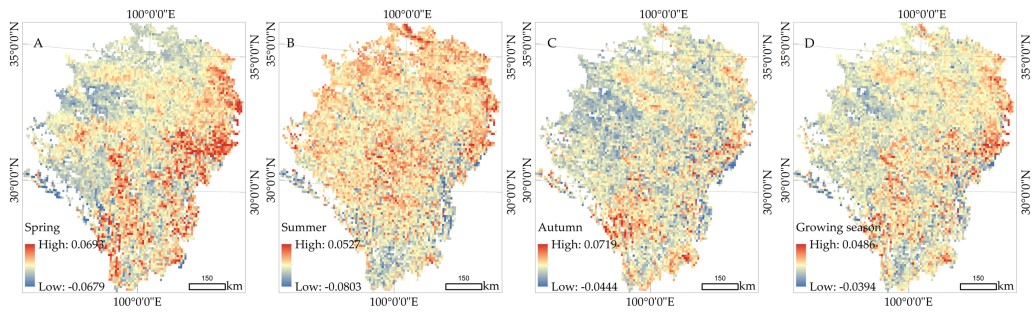

**Figure 7** **Spatial distribution of the vegetation dynamics from 1982 to 2015.** (A) Spring. (B) Summer. (C) Autumn. (D) Growing season.

decrease of NDVI was mainly distributed in the western DR region and near the dry valley in the south of the study area. The areas with obvious increases were distributed in the east section of the study area. Because the EMQTP is a sub area of QTP, the spatial distribution of vegetation change provided a quantitative value that could be compared with Peng's results (Fig. 7 of Peng's work). Although the time ranges of NDVI were greatly different (1982-2003 and 1982-2015), they can better reflect the decreasing trend of vegetation in the arid valley.

## VGT\PROBA-V NDVI-based vegetation dynamic trend from 1999 to 2018

### Phenological trends of vegetation

We used the VGT\PROBA-V NDVI with a fine resolution to calculate phenological trends. The change detection of the SGS and EGS were performed on the most recent 19 years. The trend of SGS change for the forest ecosystem showed delay ($slope = 0.66$ days/decade, $-1.96$ $<Sig. <1.96$), and the EGS showed a slight advanced trend ($slope = 2.99$ day/decade, UF $<0, -1.96 <Sig. <1.96$). Thus, it was calculated that the LGS decreased by 3.65 days/decade during this period. The SGS and EGS curves for the grassland ecosystem are shown in Figs. 8C and 8G. The SGS of the grassland ecosystem was delayed by 0.92 days/decade, and the EGS was delayed by 1.05 days/ decade. Accordingly, the LGS was prolonged by 0.03 days/decade. Compared with the forest ecosystem, the grassland ecosystem showed an obvious uptrend. In the farmland ecosystem (Figs. 8A and 8D), the SGS was delayed by 5.76 days/decade and the EGS was delayed by 0.67 days/decade. Due to the delays of the SGS and the EGS, the LGS was shortened by 5.09 days/decade.

### Spatial dynamics of vegetation

In the most recent 19 years (1999-2018), NDVI showed a decreasing trend with the area accounting for 63.4% in spring (Fig. 9A). Most was grassland cover area with a slope less than 0.004/decade. Moreover, NDVI had decreased significantly in the eastern and southern dry valleys of the study area. This area was covered by forest, especially obvious in the east of REG, south MEK, and the northwestern section of KD. The areas of NDVI increase in summer (Fig. 9B), autumn (Fig. 9C), and the growing season (Fig. 9D) were 71.8%, 60.9% and 65.4%, respectively. The slopes of NDVI increase and decrease were similar and the

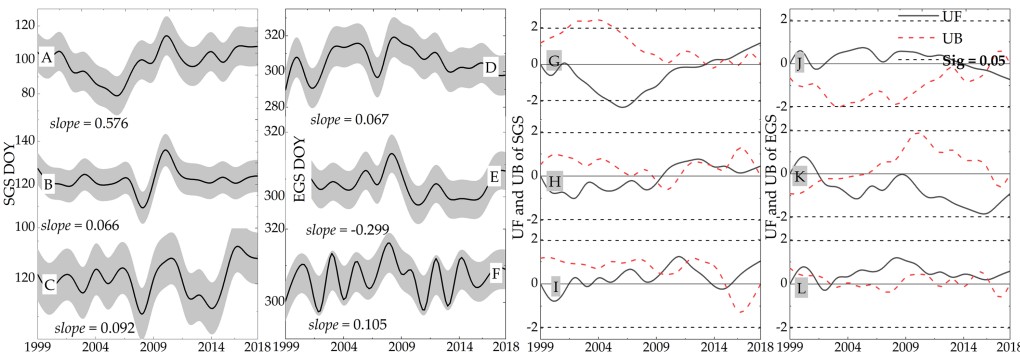

**Figure 8** **Spatial distribution of the vegetation dynamics from 1999 to 2015.** (A) Spring. (B) Summer. (C) Autumn. (D) Growing season. (E) Spring. (F) Summer. (G) Autumn. (H) Growing season.

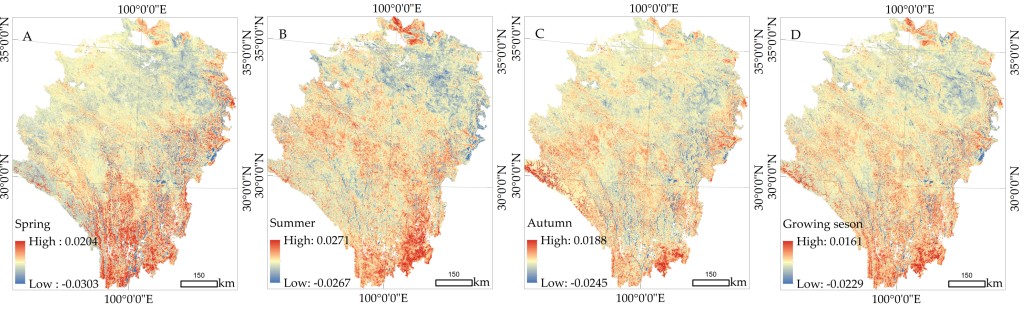

**Figure 9** **Spatial distribution of the vegetation dynamics in different seasons from 1999 to 2018.** (A) Spring. (B) Summer. (C) Autumn. (D) Growing season.

distribution area was roughly consistent. Compared with the spatial distribution of NDVI changes from 1982 to 2015, a large area near REG showed some obvious differences. This area was covered by alpine and sub-alpine meadows, and NDVI showed a downward trend with a slope of about 0.005/decade, especially in the summer (Fig. 9B). This indicated that vegetation degradation in the NEG region has been more noticeable in the last 19 years, and more attention should be paid to the changes in the ecological environment of this area.

## Crossing analysis of the vegetation dynamic from 1999 to 2015

We used GIMMS 3g NDVI and VGT\PROBA-V NDVI to perform trend analysis across different seasons between 1999 and 2015. The vegetation dynamics reflected the change trend and spatial distribution of vegetation. We divided the vegetation dynamics into three categories (decrease, weak change, and increase) according to the trend ratio of vegetation, ultimately creating the vegetation dynamic map (Fig. 10). GIMMS 3g NDVI's general change trend was basically consistent with the results that used VGT\PROBA-V NDVI. Evidently, the spatial distribution of vegetation dynamics using VGT\PROBA-V NDVI appeared finer than the results using GIMMS 3g NDVI. Vegetation degradation areas were mainly distributed in the REG grassland region and the arid valley regions of the Jinsha

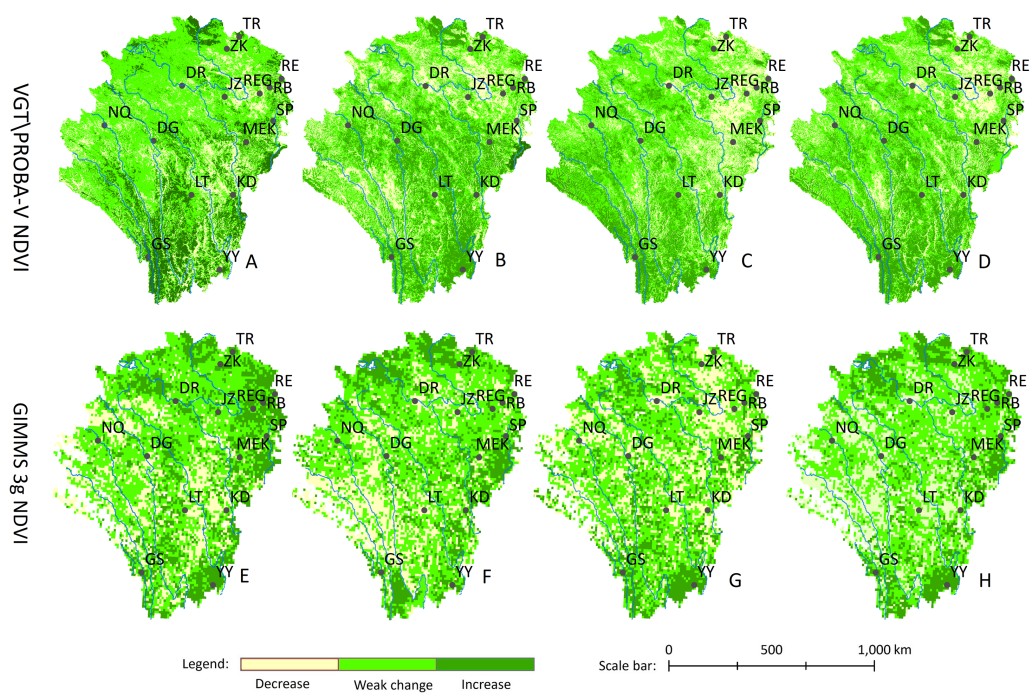

**Figure 10  Spatial distribution of the vegetation dynamics from 1999 to 2015.** (A) Spring. (B) Summer. (C) Autumn. (D) Growing season. (E) Spring. (F) Summer. (G) Autumn. (H) Growing season.

River and Yalong River. The areas with high vegetation activity appeared in the southeast of MEK and the south section of the study area, which included GS and YY.

In order to more deeply analyze the vegetation dynamic, we calculated the trend of each vegetation type at a site scale (Fig. 11). As can be seen in Figs. 11A–11H, most of the vegetation showed an upward trend, while the AM (DR, NQ and JZ site) showed a downward trend. This result was consistent with the vegetation dynamics shown in Figs. 11A and 11B. In addition, there were some differences between the two data analysis results, such as the change trend of TM vegetation, caused by the spatial resolution of the two datasets. In autumn, WL (MEK site), AM (DR, NQ and JZ site), and PG (REG site) showed a downward trend with different slopes. The downward trend of PG (REG site) was the most significant, and was also reflected in the vegetation dynamic distribution map (Figs. 10C and 10G). For the full growing season, the change trend of the seven cover types experienced an upward trend based on two NDVI image products. Compared with Peng's research results (Fig. 6 of Peng's work), the analysis at the site scale objectively reflects the NDVI change trend of the local area. It represents the change trend of various vegetation changing with the environment, rather than just a simple average of the different regions.

# DISCUSSION

## Validation of the phenological changes

We used the M-K method to achieve the general trend of phenological changes in sub section 3.3.1. In order to better detect the change trend across different phases, Bayesian

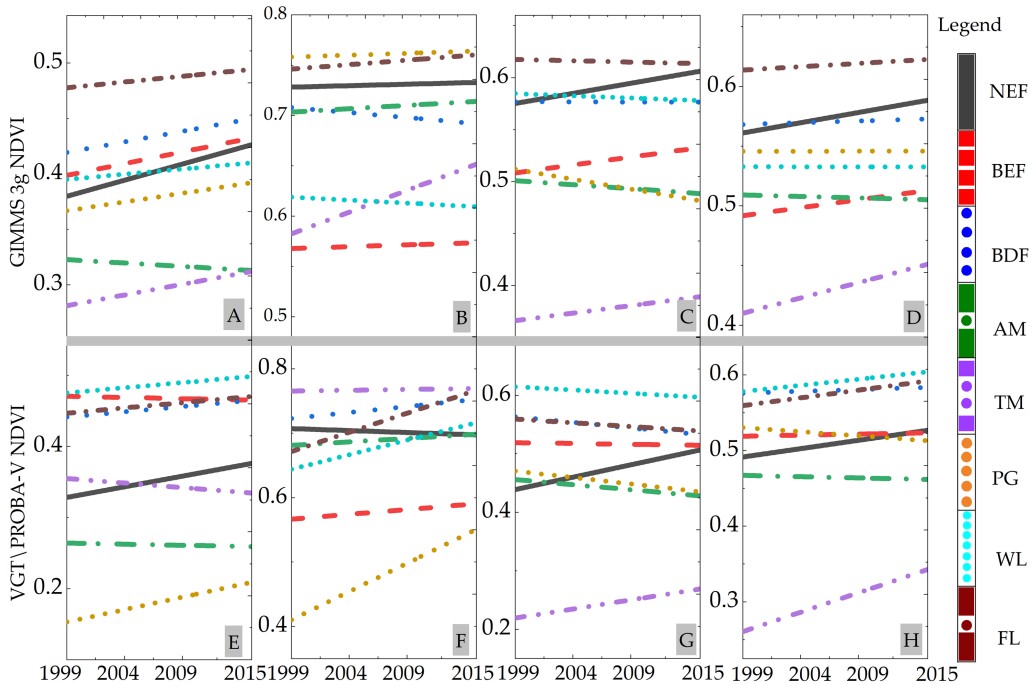

**Figure 11** **Trend of different vegetation at a site scale from 1999 to 2015.** (A) Spring. (B) Summer. (C) Autumn. (D) Growing season. (E) Spring. (F) Summer. (G) Autumn. (H) Growing season.

trend analysis was performed to obtain continuous probability and the turning points of the trend change from 1982 to 2015. The results of our calculations are shown in Fig. 12. In regard to the change trend of the SGS, the turning point was in 1998 and the probability of occurrence was more than 0.5. The turning point of the change trend in forests (Fig. 12B) was consistent with the M-K analysis results (Fig. 6E). Moreover, the change trend of the SGS was divided into two phases: before and after the turning point in 1998. Before 1998, SGS showed a downward trend, and the value of beta occurred continuously. SGS showed an upward change trend after the turning point. However, there was neither an obvious change trend nor turning point (the higher probability) for the EGS.

We selected the SGS (A-1982, B-1998, and C-2015) and EGS images of the VIP grid (D-1982, E-1998, and F-2015) to validate the phenology change from 1982 to 2015. SGS and EGS images of VIP in 1982, 1998, and 2015 can be seen in Fig. 13. The location of the sample site (Table 1) was used to extract the phenology change of vegetation based on VIP images in this study. Then the extracted data were used to validate the phenology changes of the SGS and EGS. The validation results are shown in Fig. 14. The overall trend of phenology change of VIP was consistent with that of GIMMS. In the analysis of section 3.1, we found that the phenology change showed an advanced trend before 1998, and then appeared to delay until 2015. This turning point of phenology change was also seen in the SGS change of VIP V4.

We also analyzed previous studies on phenology change in this area and nearby regions. Before 1998, the turning point of phenology change, the SGS showed an obvious advanced

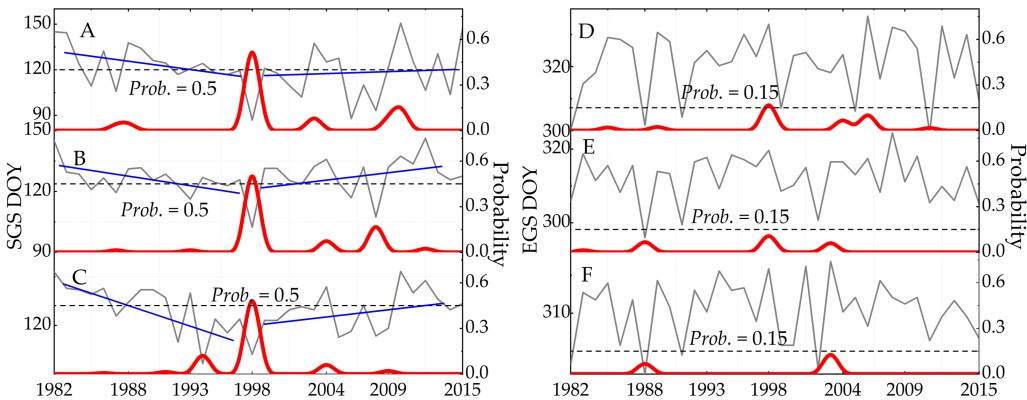

**Figure 12 The trend and the confidence level of phenologicalchanges based on the Bayesian method.** Blue solid line represents the continuous probability of trend change. Red solid line indicates the moment and probability of turning point occurring. Grey solid line denotes the change trend of raw data. (A) Farmland. (B) Forest. (C) Grassland. (D) Farmland. (E) Forest. (F) Grassland.

trend with an average slope of 10 days/decade (UF <0, −1.96 <Sig. <1.96) for various vegetation ecosystems. This was basically consistent with the results of other relevant research, which found that the slope of SGS ranged from 3.5 to 10.2 days/decade using various method and sub-areas (*Zhang, Zhang & Dong, 2013*; *Shen et al., 2014*; *Piao et al., 2011*; *Kong et al., 2017*). Many scholars have found 1998 as the approximate turning point of SGS in the QTP, which is highly consistent with the result of this study (*Piao et al., 2011*; *Shen, Piao & Dorji, 2016*). The SGS showed a delay trend after 1998, and the delays calculated by GIMMS 3g and VGT NDVI were 1.8 days/decade and 0.7 days/decade, respectively, but the significance (Sig.) was greater than 0.05 in the EMQTP. Currently, the trend of SGS change in the QTP after 1998 is a controversial topic (*Zhang, Zhang & Dong, 2013*; *Yu, Luedeling & Xu, 2010*; *Shen, Sun & Wang, 2013*). In this study, the average EGS of vegetation ecosystem was delayed by 1 day/decade between 1982 and 2018. This is consistent with Che's finding (*Che, Chen & Innes, 2014*) that the phenological change trend of the QTP at the end of the growing season was not guaranteed (*slope = 0.96 day/decade*).

## Validation of the vegetation dynamics

The analysis of section 3.2 and 3.3 show that vegetation degradation has been obvious in the REG area, especially in the last 20 years. Thus, the validation of vegetation dynamics in this area was performed based on the Landsat NDVI. Because the most significant degradation appears in autumn, we used autumn Landsat NDVI to validate the vegetation dynamics. Because of the quality (the cloud was less than 5%) of the Landsat orthophoto, the 10 scenes selected were mainly from September and October, and the time difference was less than a week. Moreover, we designed five sites to make the results of the validation more intuitive. The validation results can be seen in Fig. 15. Although the resolutions of VGT\PROBA-V NDVI (1 km) and Landsat NDVI (30 m) were different, the general vegetation dynamic trends were fundamentally consistent (Figs. 15A and 15B). In particular, the validation sites showed significant downward trends, indicating our previous calculations (sub-section

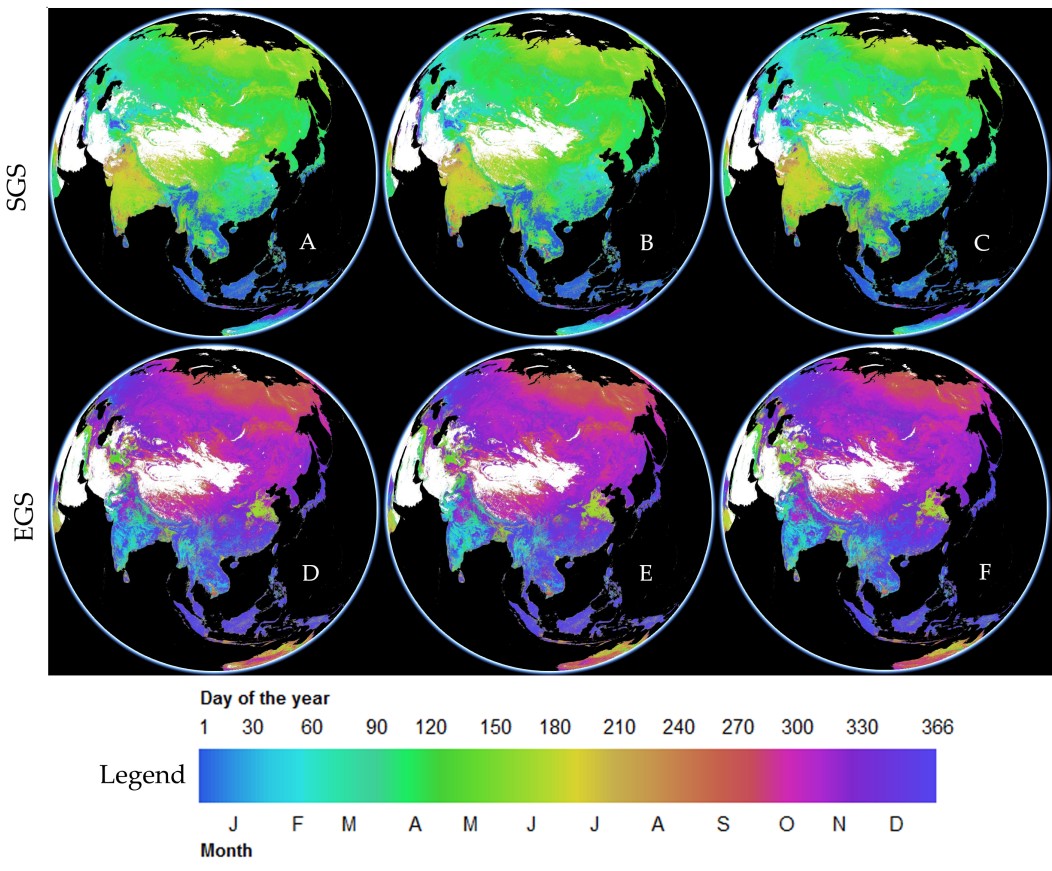

**Figure 13  Using the data engine of VIP Lab to generate the global three-dimensional phenological grid (yearly product), including the start of growing season (SGS) and end of growing season (EGS).** The data is a grid of spherical surface, and we only showed the section of the study area and surrounding areas. (A), (B), (C) Spatial distribution of SGS. (D), (E), (F) Spatial distribution of EGS.

3.2) of vegetation degradation were correct. In the circle image (the zoom view of the validation site), the vegetation dynamic is due to the degradation of the alpine meadow (Site 1 and Site 2). Additionally, human activities, such as changing farmland area (Site 4) and the expansion of residential areas (Site 3 and Site 5) also have an impact on vegetation degradation. Previous studies have revealed the degradation of alpine wetland meadows in the REG area, including the shrinkage of meadows (*Li et al., 2015*), ecological degradation (*Shen et al., 2019*), and an increase of cultivated land (*Rong et al., 2010*). The validation results in section 3 showed that vegetation dynamics and change trends are reasonable and credible. Other previous works (*Chen, 2018*; *Bian, Li & Zhang, 2017*) have also proven the vegetation degradation in REG area, and that the reason for vegetation degradation in the dry valley near the southern section isrelated to climate change (*Wang, Yao & Wortley, 2018*).

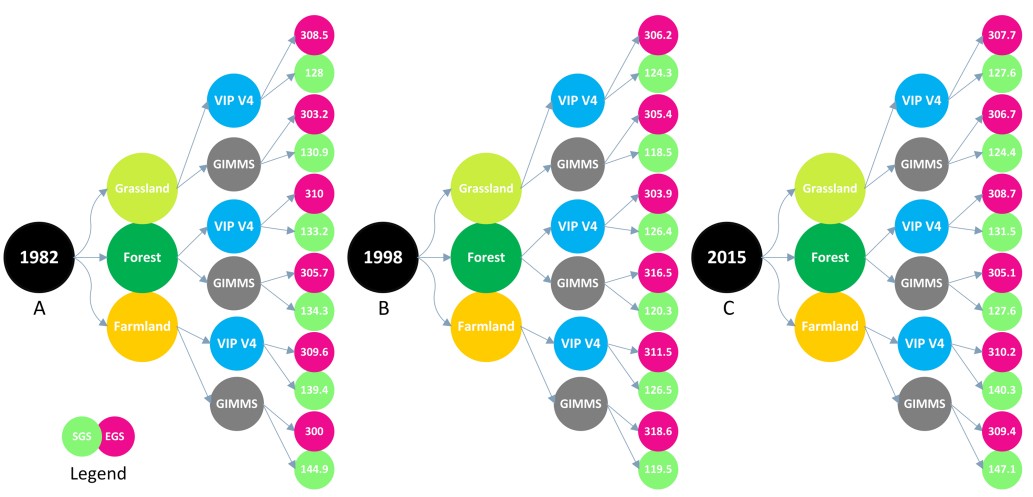

**Figure 14** SGS and EGS of various vegetations were extracted from the VIP phenological grid, and the value extracted was used to validate the GIMMS 3g NDVI-based phenological changes in specific year. (A) 1982. (B) 1998. (C) 2015.

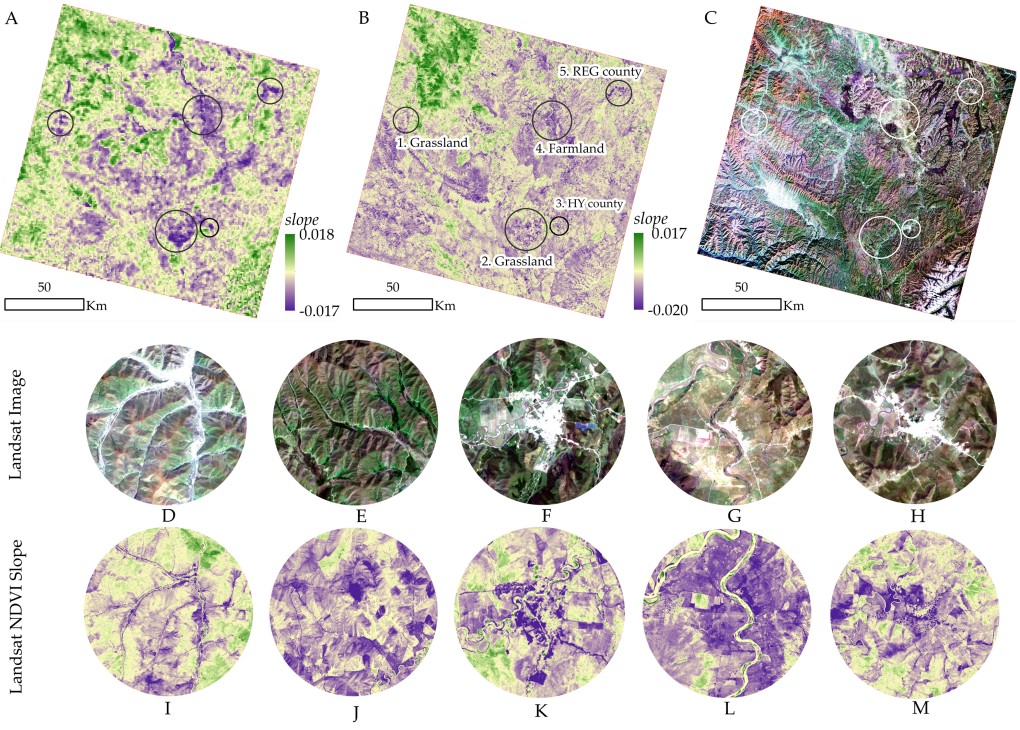

**Figure 15** Validation of the vegetation dynamics from 1999 to 2015. A, B, and C is the slope of VGT/PROBA-V NDVI, the slope of Landsat NDVI, and true color image of Landsat OLI (2015/10/01), respectively. (A) Slope of VGT NDVI. (B) Slope of Landsat NDVI. (C) Landsat 8 OLI image. (D) Site 1. (E) Site 2. (F) Site 3. (G) Site 4. (H) Site 5. (I) Site 1. (J) Site 2. (K) Site 3. (L) Site 4. (M) Site 5.

## CONCLUSIONS

We combined the high spatial resolution of Landsat NDVI with the long temporal span of GIMMS 3g NDVI and VGT/PROBA-V NDVI to establish a 37-year record of vegetation dynamics in the EMQTP at 1 km and 8 km spatial resolution. Our research focused on the phenological change of vegetation ecosystems, the trends of change, and fluctuation characteristics for various vegetation types in this sensitive area responding to global environmental change. The acquired findings on vegetation dynamics in the EMQTP, particularly in phenological changes and fluctuation characteristics for various vegetation types, can be summarized as follows:

(1). The SGS of vegetation ecosystems advanced from 1982 to 1998, and then experienced trending delays until 2018. The average advanced slope and delayed slope were 11.51 days/decade and 3.86 days/decade, respectively. The LGS of the vegetation ecosystem was prolonged with an average value of 4.24 days/decade, and the LGS of the farmland ecosystem showed the most significant prolongation trend.

(2). In the full growing season, the slope of the vegetation index (NDVI) ranged from −0.039/decade to 0.049/decade. Most of this area showed an upward trend with a percentage of 75.1%. When compared with the forest ecosystem, the grassland ecosystem showed weaker activity and the appearance of degradation, particularly in spring and autumn.

(3). In the most recent 19 years, vegetation has degenerated in REG and the arid valley region of the Jinsha River and Yalong River. This finding has also been confirmed by Landsat NDVI. Vegetation degradation is due to the shrinking of alpine meadows and the impact of human activities, which include the change of farmland area and the expansion of residential areas.

## ACKNOWLEDGEMENTS

We thank VITO, ECOCAST, Global Inventory Modeling and Mapping Studies (GIMMS), and WESTDC for the freely downloadable VGT\PROBA-V and GIMMS 3g NDVI datasets, SRTM DEM, and vegetation vector map products, respectively. We would also like to thank the anonymous reviewers for their constructive and detailed comments.

### Funding

This research was funded by the National Key R&D Program of China (NO.2017YFC0505001), the National Natural Science Foundation of China (NO. 41501060), and the Fund Project of Science & Technology Department of Sichuan Province. The funders had no role in study design, data collection and analysis, decision to publish, or preparation of the manuscript.

### Grant Disclosures

The following grant information was disclosed by the authors:
National Key R&D Program of China: 2017YFC0505001.

National Natural Science Foundation of China: 41501060.
Fund Project of Science & Technology Department of Sichuan Province.

## Competing Interests

The authors declare there are no competing interests.

## Author Contributions

- Haijun Wang conceived and designed the experiments, performed the experiments, analyzed the data, prepared figures and/or tables, authored or reviewed drafts of the paper, approved the final draft.
- Peihao Peng conceived and designed the experiments, prepared figures and/or tables, approved the final draft.
- Xiangdong Kong analyzed the data, prepared figures and/or tables, approved the final draft.
- Tingbin Zhang performed the experiments, contributed reagents/materials/analysis tools, authored or reviewed drafts of the paper, approved the final draft.
- Guihua Yi conceived and designed the experiments, contributed reagents/materials/-analysis tools, prepared figures and/or tables, approved the final draft.

## Data Availability

The raw data are available in the Supplemental Files.

## Supplemental Information

Supplemental information for this article can be found online at http://dx.doi.org/10.7717/peerj.8223#supplemental-information.

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
