# Peer review of "Vegetation dynamic analysis based on multisource remote sensing data in the east margin of the Qinghai-Tibet Plateau, China"

_PeerJ, doi:10.7717/peerj.8223_

## Round 0.1 · original submission · Major Revisions

The reviewers provided a lengthy list of corrections to be made. These all need to be corrected before further assessment is made. Especially three main issues require deep and thorough reference: 1) the methodology needs to be much more clearly described; 2) the novelty in this research and especially the difference from Peng et al. (Ecological Indicators, 2012) need to be clarified "rock solid" and 3) The manuscript needs to go through English editing and proof reading by a native speaker.

[]

Reviewer 1 ·

Basic reporting

This manuscript is far from well-writing and must be improved strongly. Some sentences are too long and complex, and with grammatical errors. Please carefully check and revise (I think the revision by native speakers is essential). Please refer to the specific comments (but not limited to).

I think the general field background is provided sufficiently, but could be improved according to the revision for research motivations (See the next section).

The article structure, figures, and tables could be improved. Please see the specific comments.

Experimental design

Generally I think this research is within the aims and scope of the journal, and the technique flow applied in this research is reasonable.

The motivation to study the scientific questions were explained inadequately, and I suggest to elaborate why the eastern edge of the Qinghai-Tibetan Plateau is important and valuable to be specially studied. Besides, the reason to study the consistency of the two datasets were not mentioned.

The description of materials & methods could be more concise but must be more exact. For example, the details for extracting the phenological variables (L175-L177) should be expounded (Also see the specific comments). Besides, I think the study area should be introduced with more detailed information on the geographic range and conditions, and how the representative sites chose should be explained in the section of methods.

Validity of the findings

This research is of less novelty as a case study and the authors should clarify the importance of this study area and what the difference varies from the related research.

I think data applied in this research is robust. My main concern is that the interannual analysis should be carried out in the same valid (with vegetation) pixels for all the years. But it seems the authors directly conducted the variation and trend analysis at all pixels including non-vegetation pixels and didn’t consider the changes of land cover.

Additional comments

More specific comments:
L31 and L40. “with…was…” please rephrase.
L38. “Ruergai” should be “Ruoergai”?
L43-L44. Rephrase.
L47-48. Rephrase.
L49. “the”
L50. “vital” ->” vitally”.
L55-58. I don’t think this conclusion (GIMMS-NDVI and SPOT-VGT are the best NDVI products) could be summarized from the cited two papers.
L62-L67. I think the methods or specific indicators applied in these cited papers are less important here than the research objects, study area or results related to the Qinghai-Tibetan Plateau. And the related research backgrounds about the Qinghai-Tibetan Plateau were not summarized enough.
L68. Is there any reason for “EMQTP is an important epitome of QTP”?
L85. “long-time series” -> “long-term time series”.
L98. Please add a reference for the GIMMS dataset.
I think the numbers of the scenes of satellite images are unnecessary to be shown.
L116. Rephrase.
L115-117. I think such pre-processing, which is record in the user manual of datasets, is unnecessary to be shown.
L120. Please add references and general information (e.g. version and producer) of the software tools.
L127. Reference.
L146. “remains unchanged after interchange”. Please rephrase.
L156. Please clarify how the seasons determined in this study.
L197. Are all the pixels valid in all the years?
L206 and L209. Please add references.
L212-L215. What the variables in the wavelet variance represent in this research should be clarified.
L219-L222. Rephrase.
L277-L282. Move to the section of methods.
The section of results. I think mixing the interannual analysis and the comparison between GIMMS-NDVI and SPOT-VGT is confusing. I suggest to describe the results separately, especially there could be one specific section for data comparison.
L309. What does “case study” mean here? I think the cited papers are for the whole TP, except for the cited paper Zhang et al. (2013) which is not in the list of reference.
L310-L311. I think it’s too casual to attribute the difference to this.
L352-L353. Please add references.
Where is Fig.2 described in the manuscript?
Figure 3. What these curves represent for should be clarified, the value of one pixel or average of the study area, and one certain year or the multi-year mean. Besides, I think such comparison is not informative enough and unnecessary.
Figure 7. The caption is incomplete. The caption of Figure 8 should be rephrased.

Reviewer 2 ·

Basic reporting

The manuscript as presented is generally sound, using good English. There were some typos which should be double-checked for the entire text.
For example:
Line 57: VGEATION -> VEGETATION
Lines 167-169: Eq. (4) should be written as a mathematical equation or programming code. "v" should be NDVI (italic). "i" did not exist in Eq. (4).
Lines 221-: "p" should be italic.

Please use consistently only one abbreviation for each term such as "SPOT VGT" or "VGT" or "VGT NDVI"; "VGTPROBA-V" or "VGT\PROBA-V NDVI"; "GIMMS NDVI" or "GIMMS 3g NDVI".
Please check all figure titles by a native English speaker.

Fig. 1: Add scale bar and north arrow; Place legend of B inside its box.
Fig. 2: Add explanation for GIMMS 3g NDVI in the title.
Fig. 4: Not professional, especially the position of the text.
Fig. 6: Title was not clear. Why were c, d placed after e,f? What was each box represented? Ecosystems (If yes, please add their names)? Why 2017-2018 was not included?
Fig. 7: Add e-h into the title. Add scale bar.
Fig. 8: Add B-D into the title.
Fig. 9: "No change" in the legend but "weak change" in the title. Add scale bar.
Fig. 10: Add legend title, the x-axis, and y-axis labels.

Experimental design

The results are somehow valuable and the methodological approach will be of interest to some readers. However, I must be honest that this paper did not bring something which was really new in term of the research topic, data, the method, and even study area.
This research was very similar to Peng et al. (Ecological Indicators, 2012). Peng et al. analyzed vegetation dynamic trend during 1982–2003 by using GIMMS NDVI data for the whole area of Qinghai–Tibet Plateau which was three times as big as the study area of this manuscript. Basically, the findings might be an updated version. However, we have little idea about the differences between the two studies because the authors did not describe or compare deeply their project and Peng's project in the introduction, result, and discussion.

The good point of this research that VGT\PROBA-V NDVI was used. Because two types of NDVI data were different in term of spatial resolution and sensors, their generated results should be analyzed independently before comparison for the same period. For example, the structure of results can be presented as below.
1. GIMMS NDVI-based vegetation dynamic trend from 1982-2015
2. VGT\PROBA-V NDVI-based vegetation dynamic trend from 1998-2018
3. Comparison of GIMMS NDVI and VGT\PROBA-V NDVI data during 1998-2015
In comparison, it is great if the advance of VGT\PROBA-V NDVI data could be clarified because of the finer resolution (1km against 8km).

Some minor suggestions:
- Please clarify SPOT VEGETATION 1 or 2 was used.
- Which UTM zone was projected?
- Lines 115-117: Valid range of DN for NDVI must be 3-255. Please put the formula of DN and NDVI as a mathematical equation with cited reference.
- A Bayesian test should be used instead of p-value and t-Text because the p-value was very easy to pass. Please check [doi:10.1038/519009f] and newer related publication.

Validity of the findings

Because the r-squared was very low in the slope analysis, was vegetation trends validity and believable? It is great if the reason for vegetation dynamics could be somehow clarified. We can only guess based on the information presented here. Were vegetation dynamics linked to climate change or deforestation or land-use change?

Additional comments

Please consider my comments and suggestions to improve the quality of the manuscript.

---

## Round 0.2 · Minor Revisions

Please see reviewer 1 question and provide an answer.

Also please see reviewer 2 comments and correct. Note that the ms needs English proof-reading.

Reviewer 1 ·

Basic reporting

'no comment'

Experimental design

'no comment'

Validity of the findings

'no comment'

Additional comments

where is the Bayesian approach?

Reviewer 2 ·

Basic reporting

The manuscript was much improved; however, it needs to be checked by a native English speaker. There are still grammar mistakes, for example, L50-55, L285-286, etc.

Figure 10 should be introduced earlier, in the section of image sources.
Revise the captions of Figs.10 and 11 to help readers understand the contents of these figures. Many readers will read the figures separately from the text.

Table 1: I can't distinguish the dates of each sensor.

Experimental design

Because EMQTP is part of that of Peng’s work and Peng also identified the vegetation dynamic within the area, the originality of the research should be clarified more clearly in the introduction section. The author mentioned in L64 that similar works were already done, so why does this research need to be conducted? Please give readers strong reasons and new points.

Because four types of remote sensing images were used, an image coregistration technique should be added in the method section.

Validity of the findings

Again, the authors should compare their results to previous works. Are there any different phenomena?

L267: the authors conclude that the result of GIMMS 3g NDVI-based was basically consistent with that of VGT\PROBA-V NDVI-based. What does this mean? Does higher resolution NDVI data not bring any benefits? Please add more discussion.

Additional comments

Other considerations:
- What is "global change" in the first line of the abstract? Please check your scientific terms throughout the text.
- L219: should be in the method section.
-L324: Landsat was used for validation, not for mapping by combination with other data.

---

## Round 0.3 · accepted · Accept

The paper is improved now and can be accepted